# Growth and Toxigenicity of *A. flavus* on Resistant and Susceptible Peanut Genotypes

**DOI:** 10.3390/toxins14080536

**Published:** 2022-08-05

**Authors:** Theophilus Kwabla Tengey, Frederick Kankam, Dominic Ngagmayan Ndela, Daniel Frempong, William Ofori Appaw

**Affiliations:** 1Council for Scientific and Industrial Research-Savanna Agricultural Research Institute (CSIR-SARI), Nyankpala NL-1032-0471, Ghana; 2Department of Crop Science, Faculty of Agriculture, Food and Consumer Sciences, University for Development Studies, Nyankpala NL-1029-6240, Ghana; 3Department of Food Science and Technology, College of Science, Kwame Nkrumah University of Science and Technology, Kumasi AK-448-1125, Ghana

**Keywords:** aflatoxigenic, in vitro seed colonisation, non-aflatoxigenic, host plant resistance and susceptible

## Abstract

Aflatoxin contamination poses serious health concerns to consumers of peanut and peanut products. This study aimed at investigating the response of peanuts to *Aspergillus flavus* infection and aflatoxin accumulation. Isolates of *A. flavus* were characterised either as aflatoxigenic or non-aflatoxigenic using multiple cultural techniques. The selected isolates were used in an in vitro seed colonisation (IVSC) experiment on two *A. flavus*-resistant and susceptible peanut genotypes. Disease incidence, severity, and aflatoxin accumulation were measured. Genotypes differed significantly (*p* < 0.001) in terms of the incidence and severity of aflatoxigenic and non-aflatoxigenic *A. flavus* infection with the non-aflatoxigenic isolate having significantly higher incidence and severity values. There was no accumulation of aflatoxins in peanut genotypes inoculated with non-aflatoxigenic isolate, indicating its potential as a biocontrol agent. Inoculations with the aflatoxigenic isolate resulted in the accumulation of aflatoxin B_1_ and G_1_ in all the peanut genotypes. Aflatoxin B_2_ was not detected in ICGV–03401 (resistant genotype), while it was present and higher in Manipinta (susceptible genotype) than L027B (resistant genotype). ICGV–03401 can resist fungal infection and aflatoxin accumulation than L027B and Manipinta. Non-aflatoxigenic isolate detected in this study could further be investigated as a biocontrol agent.

## 1. Introduction

Peanut (*Arachis hypogaea* L.) is an excellent source of plant nutrient mostly cultivated in tropical and sub-tropical regions [1,2,3]. Peanut originates from Latin America and was introduced in the 16th century into the African continent by the Portuguese [4,5]. Peanut production serves as a source of livelihood for farmers [6,7]. It is a source of vegetable oil (40%–60%), protein (20%–40%), carbohydrate content (10%–13%), numerous vitamins and minerals, haulm, and cake as food for humans and feed for livestock [1,5,8].

Peanut production is faced with challenges such as inadequate inputs, unreliable rains, the use of unimproved varieties, and disease and pest infestation [9]. In addition to the above, peanut is reported to be prone to *Aspergillus* species infection [10,11,12], which leads to a decline in its quality and quantity [13,14]. Among the *Aspergillus* species, *A. flavus* and *A. parasiticus* are the major contributors to aflatoxin production at the pre and postharvest stages, although *A. flavus* is reported to be the most prevalent in Ghana [14,15]. Aflatoxin contamination is a food safety concern due to its detrimental health impact and its resulting economic loss to peanut growers and dealers [16]. Aflatoxin contamination can also result in the reduction in the quality of grain and oilseed and depletion in their nutritional value [11].

Research discoveries have established that *A. flavus* is the most dominant and the most frequently isolated species from peanut seeds, products, and farmers’ fields leading to aflatoxin contaminations [13,17,18,19,20]. However, due to gene mutations in the aflatoxin biosynthesis pathway, not all *A. flavus* isolates are aflatoxigenic [13]. The *Aspergillus section flavi* are grouped into aflatoxin-producing and non-aflatoxin producing species, which have similarities in their morphological characteristics [21,22].

In order to deploy and use host plant resistance, *A. flavus* and aflatoxin genetics must be well understood to develop effective methods of identifying genes of resistance and the availability of low-cost aflatoxin assays to correctly verify aflatoxin accumulation is needed [23].

Resistance to *A. flavus* infection and aflatoxin accumulation is quantitative in nature [24,25], and different mechanisms probably contribute to resistance in different peanut species, possibly under different environmental conditions [26]. Resistance may result from prevention of fungal infection on the peanut seed, prevention of growth of the fungus once the infection has occurred, inhibition of aflatoxin production following infection;, and degradation of aflatoxin by products or enzymes produced by the plant or by the fungus itself [27,28].

This study sought to culturally characterise *A. flavus* isolates based on their aflatoxigenicity and distinguish resistance mechanisms acting on resistant and susceptible peanut genotypes following inoculations with aflatoxigenic and non-aflatoxigenic *A. flavus* in vitro.

Measurements of fungal incidence, severity, and aflatoxin accumulations in the *A. flavus* resistant peanut genotypes (ICGV–03401 and L027B) and susceptible peanut genotype (Manipinta) will help to distinguish the different resistance mechanisms acting in them.

## 2. Results

### 2.1. Characterisation of A. Flavus into Aflatoxigenic and Non-Aflatoxigenic Isolates

*A. flavus* isolates cultured on PDA and YESA+β-cyclodextrin media were observed through UV light at 312 nm as shown in Figure 1.

Additionally, the isolates of *A. flavus* on PDA in each Petri dish were exposed to 2 mL of concentrated Ammonia solution for 4 to 5 min (Figure 2).

The results revealed that, on the PDA, 62.5% were characterised as aflatoxigenic whilst 37% were non-aflatoxigenic *A. flavus*. Similarly, on the YESA+β-cyclodextrin, 81.25% were aflatoxigenic while 18.75% were non-aflatoxigenic (Table 1). Additionally, it was observed that 13 out of 16 *A. flavus* isolates, which represents 81.25%, were aflatoxigenic while 3 isolates (18.75%) were non-aflatoxigenic when *A. flavus* culture plates were exposed to concentrated ammonia solution (Table 1).

### 2.2. Morphological Variation in the Growth of Aflatoxigenic and Non-Aflatoxigenic A. flavus

The results on the radial growth of aflatoxigenic isolate 8 (A8) and isolate 2 (A2), and non-aflatoxigenic *A. flavus* isolate 11 (NA11) and isolate 12 (NA12) revealed that *A. flavus* increased in diameter on the PDA as the days progressed, as shown in (Figure 3). The radial growth of the non-aflatoxigenic *A. flavus* isolates were higher than the aflatoxigenic *A. flavus* isolates. Additionally, the area under the disease progress curve (AUDPC) values of aflatoxigenic *A. flavus* isolates (A2 and A8) were significantly lower than AUDPC values for non-aflatoxigenic *A. flavus* isolates (NA11 and NA12) during the 8-day incubation period (Figure 4).

### 2.3. Incidence of Aflatoxigenic and Non-Aflatoxigenic A. flavus

There was highly significant (*p* < 0.001) variation in incidence of the three peanut genotypes (ICGV–03401, L027B, and Manipinta) inoculated with aflatoxigenic and non-aflatoxigenic *A. flavus.* The resistant genotypes (ICGV–03401 and L027B) had incidence values less than 40% for aflatoxigenic and non-aflatoxigenic *A. flavus* whereas the susceptible genotype (Manipinta) had the highest (78%–80%). However, the percentage incidence of non-aflatoxigenic *A. flavus* on all the genotypes was higher than the percentage incidence of aflatoxigenic *A. flavus* (Figure 5).

### 2.4. Severity of Aflatoxigenic and Non-Aflatoxigenic A. flavus

There were significant (*p* < 0.001) differences in severity of the three peanut genotypes inoculated with aflatoxigenic and non-aflatoxigenic *A. flavus.* The resistant genotypes (ICGV–03401 and L027B) had severities of less than 30% for aflatoxigenic and non-aflatoxigenic *A. flavus* whereas the susceptible genotype (Manipinta) had the highest (53%–63%). However, the incidence of non-aflatoxigenic *A. flavus* was higher in all the peanut genotypes than aflatoxigenic *A. flavus* (Figure 6).

### 2.5. Quantification of Aflatoxin

Extraction and estimation of aflatoxin accumulation in aflatoxigenic and non-aflatoxigenic isolates in peanut was quantified using HPLC (Figure 7) with aflatoxin standards (Figure 8). The chromatogram results revealed the peaks at retention times of about 3 min. Figure 7A,C confirmed the detection and accumulation of aflatoxin B_1_, B_2_, and G_1_ in L027B and Manipinta, while only B_1_ and G_1_ was detected in ICGV−03402 Figure 7B following the inoculation of peanut seeds with the aflatoxigenic isolate A2. However, chromatogram in Figure 7D indicates non-detection of aflatoxins in peanut genotypes L027B, ICGV–03401, and Manipinta by non-aflatoxigenic isolate used in inoculation. Aflatoxin accumulation of the three peanut genotypes (ICGV–03401, L027B, and Manipinta) based on HPLC results is shown in Table 2.

## 3. Discussion

Several interventions have been reviewed and recommended for aflatoxin management in peanuts; among these, host plant resistance have proven to be the most effective [29]. The use of resistant genotypes if available is the best option. Apart from being less cost-effective and easy to disseminate, it does not require any special expertise in its application and it is compatible with other control methods [29,30]. An understanding of the genetics and mechanisms of resistance to aflatoxin will help efficiently breed for resistance to this trait in peanut.

In order to elucidate the mechanisms of resistance in selected *A. flavus* resistant and susceptible peanut genotypes, it was hypothesised that (1) Non-aflatoxigenic *A. flavus* grows faster than aflatoxigenic *A. flavus* and could therefore limit the accumulation of aflatoxin and (2) *A. flavus* infection and colonisation is influenced by the genetic makeup of the peanut genotype and that subsequent aflatoxin production is dependent on the toxigenicity of the pathogen. These were investigated through multiple cultural methods of determining the aflatoxigenicity of the isolates followed by a radial growth bioassay to determine which of the isolates was fast growing. An in vitro seed colonisation (IVSC) assay was then conducted with the selected non-aflatoxigenic isolate and aflatoxigenic isolate, followed by detection and quantification of aflatoxins produced and accumulated in the infected tissues.

*A. flavus* was identified by its characteristic yellow green colour and with circular conidia which had a whitish colour around the edge [31].

The present study revealed that under the UV light observations, 81.25% of the isolates were detected by YESA+β-cyclodextrin medium as aflatoxigenic, whereas on PDA medium, 62.5% of the isolates were aflatoxigenic. A similar observation was made by Mahmoud et al. [32], who detected aflatoxigenic isolates under UV light using PDA amended with Sodium Chloride and YESA and the positive isolates observed were 20% and 53.33% respectively. This gives an indication that different culture media have different abilities in detecting aflatoxigenic and non-aflatoxigenic *A. flavus* isolates.

Again, by exposing the isolates to concentrated ammonia vapour, the study revealed that 13 (81.25%) out of the 16 isolates were observed to be aflatoxigenic whereas the remaining 3 (18.75%) isolates were non-aflatoxigenic *A. flavus.* Similar observations were made by Navya et al. [13], who identified 81.58% (31 out of 38 *A. flavus* isolates) to be aflatoxigenic *A. flavus* while the remaining 7 (18.42%) were non-aflatoxigenic in *A. flavus* culture plates exposed to ammonia vapour. This trend may mean majority of *A. flavus* isolates in most areas are aflatoxigenic as reported in previous studies [33,34,35]. The percentage of aflatoxigenic strains of *A. flavus* has been demonstrated to vary with the kind of substrate and environmental conditions, according to a study by Bharose et al. [19].

The AUDPC values and morphological variation in radial growth revealed that the non-aflatoxigenic *A. flavus* isolates grows faster than the aflatoxigenic isolates. This corroborates with reports that in nature the non-aflatoxigenic *A. flavus* grows faster than aflatoxigenic *A. flavus* since it does not require energy to produce aflatoxins and can invade and displace aflatoxigenic isolates [13,35].

The peanut seeds from the three genotypes reacted differently to aflatoxigenic and non-aflatoxigenic *A. flavus* infection as shown by their incidence and severity values. These differences could be associated with the differences in genetic makeup of the genotypes. ICGV–03401 and L027B had the lowest incidence and severity values, Manipinta had the highest. Among the two putative resistant lines, ICGV–03401 highly restricted fungal infection and colonisation, which could be attributed to the thickness and permeability of its seed coat [20,29,36] or seed coat biochemicals [37].

The selected aflatoxigenic and non-aflatoxigenic isolates infected and colonised the peanut genotypes with the non-aflatoxigenic isolate, resulting in the highest percentage incidence and severity. This means that the genotypes were not immune to neither aflatoxigenic nor non-aflatoxigenic *A. flavus* infection since both strains have an equal chance to survive and multiply. The genotypes also followed the same pattern (ICGV–03401 > L027B > Manipinta) in terms of their resistance to either aflatoxigenic or non-aflatoxigenic isolates. HPLC results, however, revealed that inoculation with a non-aflatoxigenic isolate did not result in the production and accumulation of aflatoxin but aflatoxins were detected in the genotypes inoculated with the aflatoxigenic strain. Resistance mechanisms to *A. flavus* infection are the same irrespective of the toxigenicity of the isolate. In terms of aflatoxin production, a pathogen factor (aflatoxigenic) and genotype factor (susceptibility) is involved. The toxigenicity of the isolate is very important in the production of aflatoxin. Peanut genotypes that exhibit strong resistance to any of these strains through IVSC may subsequently be resistant to aflatoxin production. IVSC could therefore be regarded as a quick way of screening for resistance to *A. flavus* infection and aflatoxin accumulation.

Aflatoxin types detected and identified in the three peanut genotypes were AFB_1_, AFB_2_, and AFG_1_ with aflatoxin accumulation ranging from 0.4 ppb to 2.096 ppb. These values were obtained after 7 days incubation period following inoculation of a highly concentrated toxigenic strain. Increase in incubation period does not really influence aflatoxin concentration [38]; however, these values could increase with respect to favourable environmental factors such as temperature and water activity and storage methods, which makes the pathogen thrive. Most studies have a longer storage time and favourable environmental conditions associated with a higher aflatoxin accumulation [38,39,40].

The non-detection of AFB_2_ in ICGV–03401 further boldens the resistance of this genotype, which could probably be due to the expression of genes that interfere with the aflatoxin biosynthetic pathway. This could mean apart from the concentration of specified aflatoxins, its absence could account for the resistance in a given genotype. Additionally, scenarios where non-aflatoxigenic *A. flavus* infection did not lead to aflatoxin production could be attributed to the presence of deleterious genes in the fungi that block the aflatoxin biosynthetic pathway. It is also possible that one or more aflatoxin biosynthetic genes may be absent, as reported by Commey et al. [37] when aflatoxin biosynthetic genes were compared between a non-aflatoxigenic *A. flavus* and aflatoxigenic *A. flavus* isolates. Non aflatoxigenic isolates have been used as biocontrol agents to reduce aflatoxin contamination in peanuts by 70%–100% in Ghana [41]. Similarly, Xu et al. [42] used non-aflatoxigenic strains to reduce aflatoxin B_1_ accumulation in peanut kernels by 90%. In countries such as USA, Senegal, Nigeria, and Gambia, non-aflatoxigenic *A. flavus* are now serving as biocontrol agents to reduce aflatoxin production commercial in peanut production [43,44].

## 4. Conclusions

In conclusion, this study isolated and categorised *A*. *flavus* isolates into aflatoxin-producing isolates and non-aflatoxin-producing isolates from infected peanut genotypes using cultural techniques. Additionally, there were variations in the infection and colonisation of resistant and the susceptible peanut genotypes when inoculated with the aflatoxigenic and non-aflatoxigenic strains. The HPLC results confirmed that inoculations with the aflatoxigenic isolate resulted in the production and accumulation of aflatoxin B_1_, B_2_, and G_1_, whereas inoculation with non-aflatoxigenic *A. flavus* did not produce any aflatoxin. The most resistant genotype lacked one of the detected aflatoxins under HPLC. Therefore, aflatoxigenic and non-aflatoxigenic strains all have equal ability to infect peanut seeds, but peanut genotypes demonstrated varied resistance to these isolates and varied aflatoxin accumulation. IVSC technology can be used as a quick method to screen for resistance to *A. flavus* infection and aflatoxin accumulation. The peanut genotypes ICGV–03401 and L027B could serve as sources of donors for resistance to *A. flavus* infection and aflatoxin accumulation. Future studies could also be carried out on the utility of non-aflatoxigenic isolates as a potential source of biocontrol agents.

## 5. Materials and Methods

### 5.1. Study Area

This experiment was carried out at the Plant pathology laboratory of the CSIR-Savanna Agricultural Research Institute, Nyankpala, Ghana.

### 5.2. Media Preparation for Fungal Isolation

#### 5.2.1. Preparation of Potato Dextrose Agar (PDA)

Culturing of *A. flavus* was carried out using Potato Dextrose Agar (PDA) medium (Oxoid) according to the manufacturer’s procedures. The prepared PDA was amended with chloramphenicol to inhibit the growth of bacteria and then dispensed into the sterilised disposable Petri dishes to solidify in an antiseptic environment [45].

#### 5.2.2. Preparation of Yeast Extract Sucrose Agar (YESA)

A total amount of 40.50 g of Yeast extract sucrose agar made up of yeast extract of 4.0 g, sucrose of 20 g, KH_2_PO_4_ of 1.0 g, MgSO_4_ of 0.5 g, and agar of 15 g was used and prepared according to the manufacture’s procedures. It was amended with chloramphenicol to inhibit the growth of bacteria and beta-Cyclodextrin an aflatoxin inducing chemical respectively. This was dispensed into the sterilised Petri dishes in an antiseptic environment to solidify. Plates were exposed to Ultraviolet (UV) light to check for their fluorescence.

#### 5.2.3. Source of Infected Peanuts for *A. flavus* Isolation

Infected peanut samples were obtained from a peanut processing centre at Nyankpala, Northern Region of Ghana. The infected seeds were collected and well packaged into a well labelled brown paper bag, and kept at room temperature until the time of isolation of *A. flavus*.

#### 5.2.4. Isolation of *A. flavus* Isolates

With a modified isolation method by ref. [46], infected peanut seeds obtained were surface sterilised with 0.5% sodium hypochlorite solution for 1 min, rinsed in distilled water three times, and dried on tissue paper. Using sterilised forceps in the lamina flow hood cabinet, the surface-sterilised and dried seeds were placed onto the solidified PDA growth medium in the Petri dishes (10 seeds per plate) and incubated for 72 h at 27 °C. The plates were visually observed and any visible mycelium on the seeds characterised by green-yellowish colouration was considered as the initial isolation criterion for *A. flavus*.

#### 5.2.5. Obtaining Pure Cultures of *A. flavus*

Pure culture isolates were obtained by transferring mycelium with green-yellowish colour with sterilised inoculation needle from the previous plates and inoculating onto the centre of a fresh petri dish with PDA medium under aseptic environment in the lamina flow and incubating the plates for 10 days at 25 °C [46].

#### 5.2.6. Identification of *A. flavus*

*A. flavus* was identified by observing morphological structures such as colony growth, conidia, texture, and colour [34,47] of the isolates obtained.

#### 5.2.7. Detection of Aflatoxigenic and Non-Aflatoxigenic *A. flavus*

##### YESA Medium under UV Light

Pure *A. flavus* isolates were inoculated on YESA+β-cyclodextrin medium in three replicates for 7 days at room temperature. They were viewed under the UV light at 312 nm.

##### Exposure of Isolates on PDA to Ammonia Vapour

In 9 cm glass Petri plates, each isolate of *A. flavus* was inoculated in the centre of a solidified PDA medium and cultured for 7 to 8 days at 25 °C. The plates were subjected to 2 mL of concentrated Ammonia solution for 4 to 5 min and observed for change in colour of the under-side of colonies of aflatoxin-producing isolates. Those plates that changed colour from green-yellowish colour to orange-yellow pigmentation were classified as aflatoxigenic and those that did not change colour were noted as non-aflatoxigenic [48].

#### 5.2.8. Radial Growth of Aflatoxigenic and Non-Aflatoxigenic Isolate on PDA

Four replicates of each isolate were plated on PDA and observed for their radial growth until the isolate with the fastest growth covers the entire plate. The area under the disease progress curve (AUDPC) was computed for each isolate growing on the PDA using the radial growth results for days 2, 4, 6, and 8 using the formula as described by Shaner and Finney (1977).
(1)$$\mathrm{AUDPC}=\sum_{i}^{n-1}\left(\frac{y_{i}+y_{i+1}}{2}\right)\left(t_{i+1}-t_{i}\right)$$
where *n* is the total number of observations, *y_i_* is an assessment of a disease at the *i*th observation, and *t_i_* is time at the *i*th observation.

#### 5.2.9. Peanut Genotypes

Peanut genotypes less than 2 months old after harvest and free from any fungal contamination were used in this study. The three peanut genotypes, namely, ICGV–03401, L027B, and Manipinta, used in this study were obtained from the CSIR-Savanna Agricultural Research Institute, Nyankpala, Ghana. The ICGV–03401 have been reported to have resistance to *A. flavus* infection in vitro and similarly L027B have also been reported to have resistance to *A. flavus* infection in vitro (unpublished data). Manipinta is known to be susceptible to *A. flavus* infection in vitro.

### 5.3. In Vitro Seed Colonisation

#### 5.3.1. Experimental Design

The experiment was laid out in a Completely Randomised Design (CRD) with three replications. The treatments were the three peanut genotypes and these were inoculated first with aflatoxigenic *A. flavus* in one experiment and in another experiment inoculated with non-aflatoxigenic *A. flavus*. Each plate served as a replicate with ten healthy peanut seeds.

#### 5.3.2. Seed Sterilisation and Inoculation with Aflatoxigenic and Non-Aflatoxigenic Isolate

Preceding the inoculations, 30 peanut seeds with intact seed coats of each genotype were surface sterilised with 0.5% sodium hypochlorite and rinsed properly in sterile distilled water three times. The sterilised peanut seeds were placed in Petri dishes lined with moist Whatman No. 2 filter paper. Conidial suspensions were prepared from a 10-day-old culture of *A. flavus* isolate grown on PDA medium. Each seed was inoculated using 60 µL of suspension of *A. flavus* containing 1 × 10^6^ spores per mL. They were incubated at room temperature for eight days [20,36].

After 8 days, the seeds were visually observed for seed infection by *A. flavus* by recording the percentage of seeds infested, which was shown by the presence of sporulating surface growth. Inoculations were performed using aflatoxigenic *A. flavus* and non-aflatoxigenic *A. flavus*.

#### 5.3.3. Data Collection

Data were recorded on percent incidence and severity of aflatoxigenic and non-aflatoxigenic *A. flavus*. The incidence percent was calculated using the formulae below:(2)$$\;\;\;\;\;\;\;\;\;\mathrm{Pathogen}\;\mathrm{incidence}\;=\frac{\;\mathrm{No}.\;\mathrm{of}\;\mathrm{seeds}\;\mathrm{showing}\;\mathrm{the}\;Aspergillus\;flavus\;\mathrm{Colonisation}}{\mathrm{Total}\;\mathrm{number}\;\mathrm{of}\;\mathrm{seeds}}\times 100$$

Severity data was estimated using a modified scale of 0–5 [20]. The criterion for severity estimation is defined as follows: 0: non-infected peanut seeds (Highly resistant), 1: Less than 20% peanut seed surface covered (resistant), 2: 20% to 40% peanut seed surface covered (moderately resistant), 3: 40% to 60% peanut seed surface covered (susceptible), 4: 60% to 80% peanut seed surface covered (moderately susceptible), and 5: 80% to 100% peanut seed surface covered (Highly susceptible).
(3)$$\mathrm{Pathogen}\;\mathrm{severity}\;=\frac{\;\mathrm{Sum}\;\mathrm{of}\;\mathrm{pathogen}\;\mathrm{rating}}{\mathrm{Highest}\;\mathrm{rating}\times \mathrm{Total}\;\mathrm{number}\;\mathrm{of}\;\mathrm{kernel}\;\mathrm{rated}}\times 100$$

#### 5.3.4. Detection of Aflatoxins Using High Performance Liquid Chromatography

##### Sample Extraction

Aflatoxin was extracted using methods described by ref. [49] with slight modifications of using acetonitrile: acetic acid *v*/*v* (9:1) as the extraction solution. Using a Preethi Mixer Grinder (Tamil Nadu, India), peanut samples were homogenised. A weight of 2 g of the homogenised sample was quantitatively transferred into a 15 mL centrifuge tube, 5 mL of distilled water was added, and the tube was vortexed for 1 min. After allowing the sample solution to stand for 5 min, 5 mL of the extraction solution was added. Using the Genie Vortex machine (New York, NY, USA), the resultant mixture was vortexed for 3 min. Next, 1.32 g of anhydrous MgSO_4_ and 0.2 g of NaCl were added to the mixture and vortexed for 1 min. This was followed by centrifuging the tubes for 5 min at 3300× *g*. The upper organic layer was then filtered through a 0.45 µm nylon syringe prior to injection. Finally, 50 µL of the filtered extract was injected into the high-performance liquid chromatographer (HPLC).

##### HPLC Analysis Technique

HPLC analysis was carried out based on AOAC Official Method 2005.08 (AOAC,2006) with Photochemical Reactor for Enhanced Detection (PHRED) for post-column derivatisation. A Cecil-Adept Binary Pump HPLC (Cambridge, UK) was joined with Shimadzu 10AxL fluorescence detector (Shimadzu Corporation, Tokyo, Japan) (Ex: 360 nm, Em: 440 nm) with Sunfire^®^ C18 Column (150 × 4.60 mm, 5 um). The mobile phase used was methanol: water (40:60, *v*/*v*) at a flow rate of 1 mL/min with column temperature maintained at 40 °C. LCTech UVE (Dorfen, Germany) was used for post-colum photochemical derivatisation. The aflatoxin mix (G_1_, G_2_, B_1_, B_2_) standards (ng/g) of 5.02 ng/µL in acetonitrile from Romer Labs^®^ was used for matrix-based calibration. Aflatoxins in the samples were detected by using the retentions of the standard solution run and quantification was conducted using the calibration of curves of each respective toxin. Quality assurance for established by checking for precision and trueness by spiking blank samples with aflatoxins standard (Table 3). Blank samples were run periodically confined to the absence of aflatoxins. The coefficient of variation obtained for replicates was less than 15%.

#### 5.3.5. Data Analysis

Data were subjected to analysis of variance using GenStat (12th Edition) version 12.0.0.3033. Treatment means were separated using the Least Significance Difference (LSD) at 5% significant level.

## Figures and Tables

**Figure 1 toxins-14-00536-f001:**
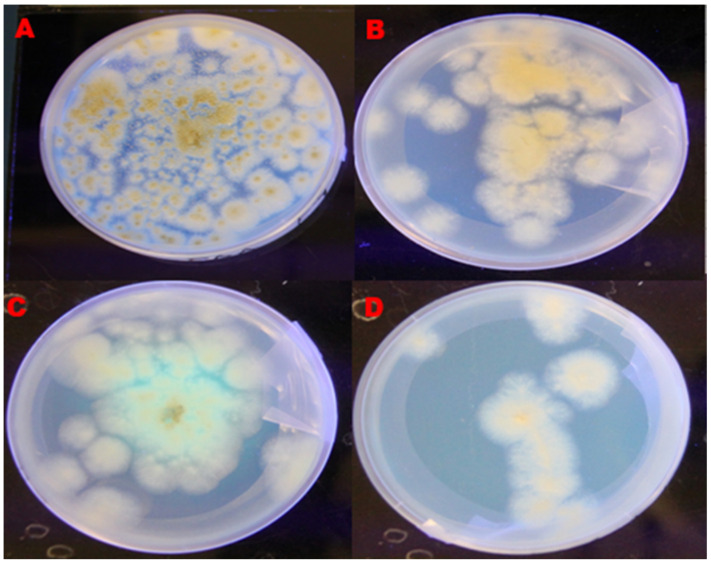
*A. flavus* colony observed under UV light at 312 nm. (**A**) *A. flavus* isolate grown on PDA fluoresces under UV light indicating aflatoxigenicity. (**B**) *A. flavus* isolate grown on PDA did not fluoresce under UV light indicating, non-aflatoxigenicity. (**C**) *A. flavus* isolate grown on YESA produced a blue fluorescence under UV light, indicating aflatoxigenicity and (**D**) *A. flavus* isolate grown on YESA did not produce a blue fluorescence under UV light, indicating non-aflatoxigenicity.

**Figure 2 toxins-14-00536-f002:**
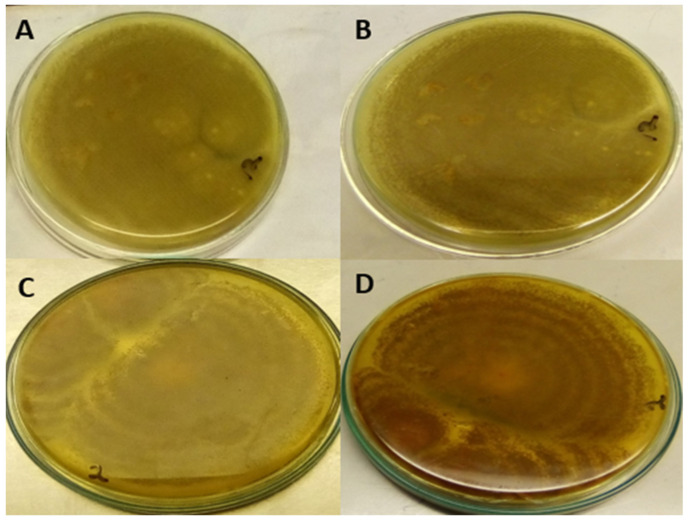
Colour of the underside of *A. flavus* colony on PDA when exposed to ammonia solution. (**A**) Non-aflatoxigenic *A. flavus* isolate before exposure and (**B**) non-aflatoxigenic isolate after exposure. (**C**) Aflatoxigenic isolate of *A. flavus* before exposure to Ammonia solution and (**D**) aflatoxigenic isolate after exposure to ammonia solution.

**Figure 3 toxins-14-00536-f003:**
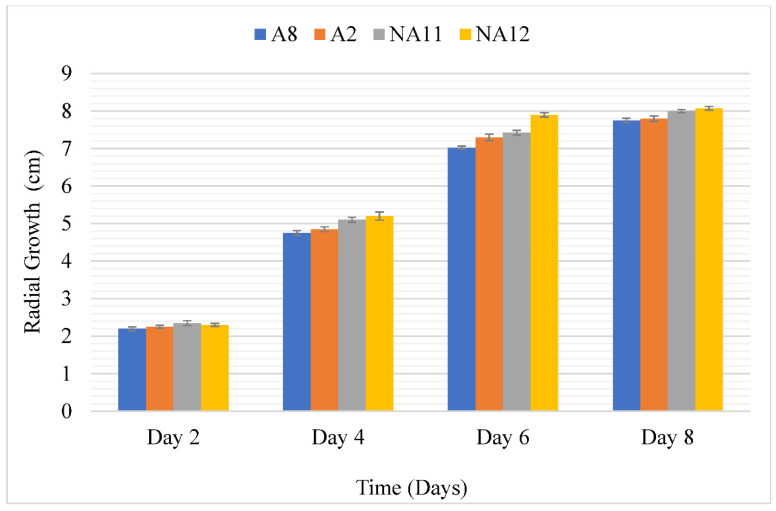
Radial growth of aflatoxigenic (A) and non-aflatoxigenic (NA) *A. flavus* on PDA. Error bars represent standard errors of means. A2—Aflatoxigenic isolate 2, A8—Aflatoxigenic isolate 8, NA11—Non-aflatoxigenic isolate 11, NA12—Non-aflatoxigenic isolate 12.

**Figure 4 toxins-14-00536-f004:**
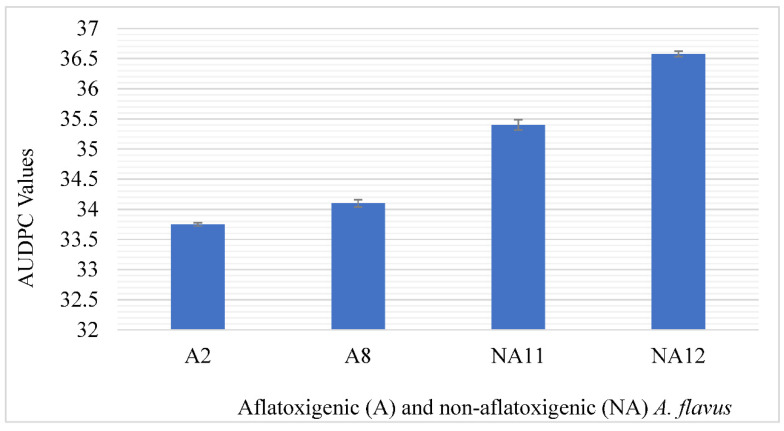
Area under disease progress curve (AUDPC) value for aflatoxigenic and non-aflatoxigenic *A. flavus* isolates on PDA. Error bars represent standard errors of means. A2—Aflatoxigenic isolate 2, A8—Aflatoxigenic isolate 8, NA11—Non-aflatoxigenic isolate 11, NA12—Non-aflatoxigenic isolate 12.

**Figure 5 toxins-14-00536-f005:**
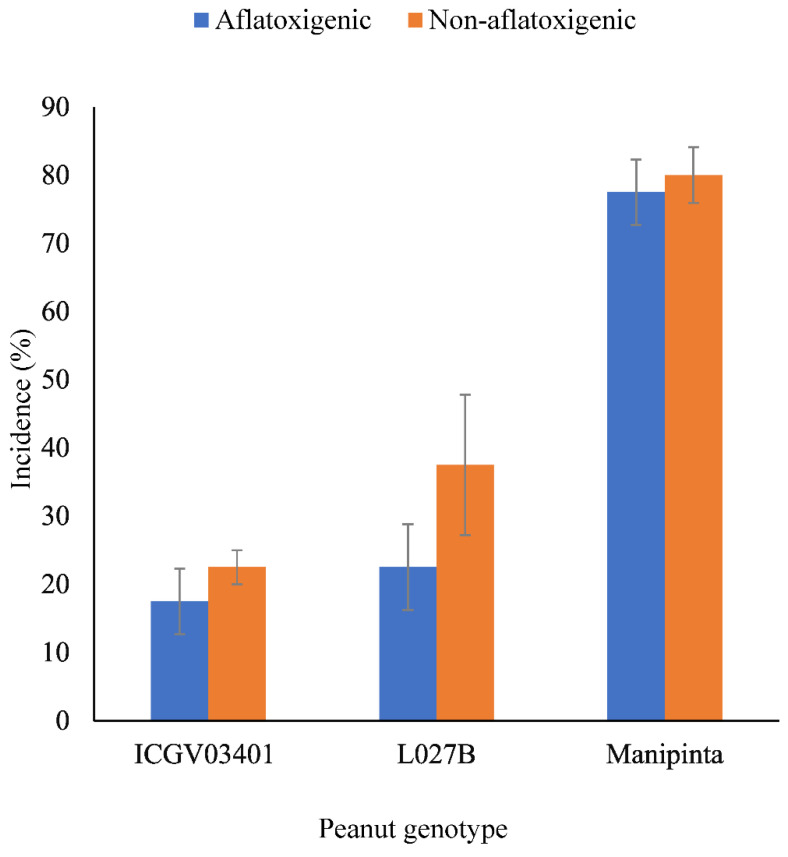
Incidence of aflatoxigenic and non-aflatoxigenic *A. flavus* on resistant peanut genotypes (ICGV–03401 and L027B) and susceptible peanut genotype (Manipinta). Error bars represent standard errors of means.

**Figure 6 toxins-14-00536-f006:**
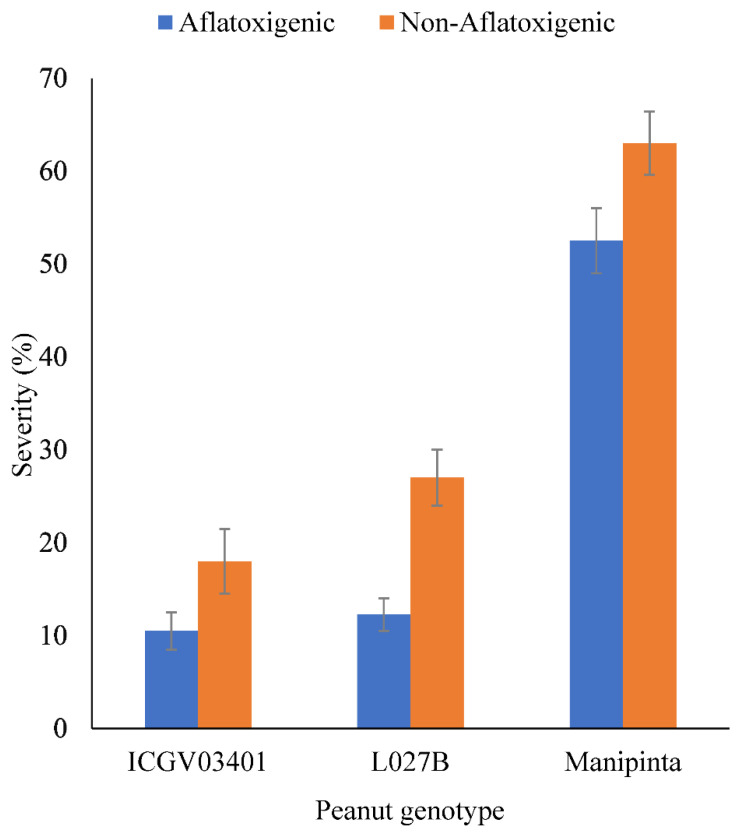
Severity of aflatoxigenic and non-aflatoxigenic *A. flavus* on resistant peanut genotypes (ICGV–03401 and L027B) and susceptible peanut genotype (Manipinta). Error bars represent standard errors of means.

**Figure 7 toxins-14-00536-f007:**
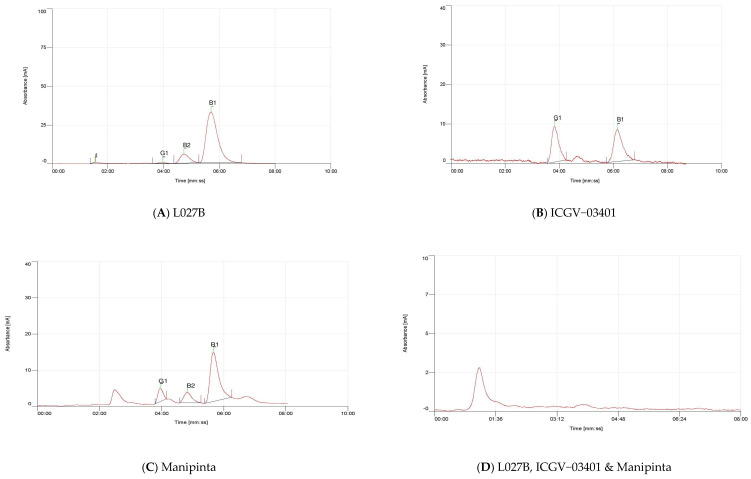
Chromatogram (**A**) shows the peaks of aflatoxin B_1_, B_2_, and G_1_ in peanut genotype L027B, chromatogram (**B**) indicates the peaks of aflatoxin B_1_ and G_1_ in ICGV–03401, chromatogram (**C**) represents the peaks of aflatoxin B_1_, B_2_, and G_1_ in Manipinta infested with aflatoxigenic isolate. The chromatogram (**D**) shows the non-detection of aflatoxin accumulation in peanut genotypes L027B, ICGV–03401, and Manipinta infested with non-aflatoxigenic isolate. Red line denotes the peaks, while black line refers to the baseline.

**Figure 8 toxins-14-00536-f008:**
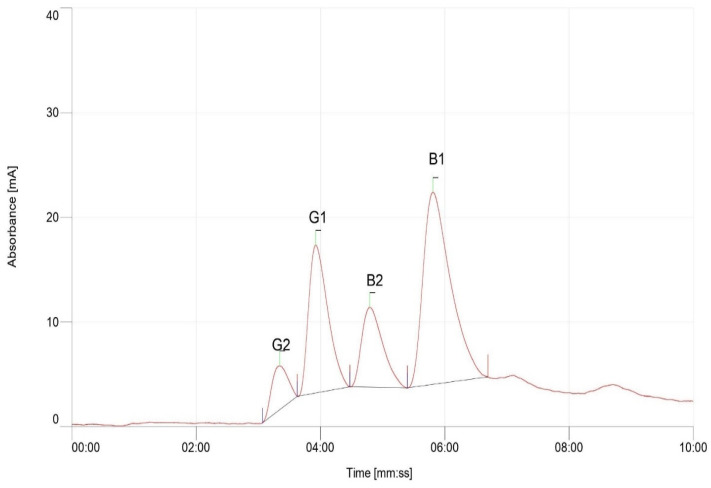
Chromatogram represents the aflatoxin standards used in the HPLC analysis. G_1_—Aflatoxin G_1_, G_2_—Aflatoxin G_2_, B_1_—Aflatoxin B_1_ and B_2_—Aflatoxin B_2_. Red line denotes the peaks, while black line refers to the baseline.

**Table 1 toxins-14-00536-t001:** Screening aflatoxigenic and non-aflatoxigenic *A. flavus* using cultural methods.

Isolates Code	UV Fluorescence		Concentrated Ammonia Solution
	PDA	YESA+β-CD	PDA
AF01	+	+	+
AF02	+	+	+
AF03	+	−	+
AF04	+	+	+
AF05	+	+	+
AF06	+	+	+
AF07	+	+	+
AF08	+	+	+
AF09	−	+	+
AF10	−	+	+
AF11	−	+	−
AF12	−	+	−
AF13	−	−	−
AF14	−	−	+
AF15	+	+	+
AF16	+	+	+

AF—*Aspergillus flavus*.

**Table 2 toxins-14-00536-t002:** Concentration of accumulated aflatoxin following in vitro inoculation with the aflatoxigenic isolate A2 on peanut genotypes.

Genotype	Concentration of Aflatoxin (ppb)		
	AFB_1_	AFB_2_	G_1_
ICGV–03401	0.86	Nil	0.76
L027B	2.10	0.85	0.40
Manipinta	0.61	1.09	0.65

**Table 3 toxins-14-00536-t003:** Quality assurance for aflatoxin analysis.

Aflatoxins	LOD (ppb)	LOQ (ppb)	R^2^	Recovery (%)
Aflatoxin B_1_	0.2	0.4	0.999	98 ± 0.71
Aflatoxin B_2_	0.1	0.2	0.999	98 ± 1.05
Aflatoxin G_1_	0.2	0.4	0.999	99 ± 0.14
Aflatoxin G_2_	0.1	0.2	0.995	99 ± 0.62

Aflatoxin calculation; Aflatoxin (ng/g) = $A\;\times \frac{T}{I}\times \frac{1}{W}$, where A = ng of aflatoxin as eluate injected, T = volume (µL) of eluate of the final test solution, I = volume eluate injected into LC (µL), W = mass (g) of commodity (final extract).

## Data Availability

Not applicable.

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
