# Peer review of "Growth and Toxigenicity of A. flavus on Resistant and Susceptible Peanut Genotypes"

_toxins, 2022, doi:10.3390/toxins14080536_

Round 1

Reviewer 1 Report

Dear Authors,

It was a good exercise. However, it must be improved considerably.

I have many doubts about some of the conclusions that you have made according with the techniques that were used. I am not happy with the identification process, the relevance of using different cultural techniques for aflatoxin detection having an HPLC available and so on. You also have to review your reference system.

I have focused in some of these points in the file that I am sending you. Please take this into account to improve your work.

Regards

Author Response

Response to Reviewer 1 Comments

Point 1: Introduction not well written.

Response 1: It has now been modified.

Point 2: I would have used malt extract agar

Response 2: You are right with the malt extract agar. However, PDA is also suitable for growth of A. flavus.

Point 3: Figure 1 has poor quality, I do not think it is necessary.

Response 3: This figure has been removed.

Point 4: If you were able to analyse the aflatoxins in peanuts using HPLC why use this method.

Response 4: Use of HPLC is much more expensive. So for a quick detection of aflatoxigenic strain from the many A. flavus isolates it is better to use the medium amended with beta-cyclodextrin followed by detection under UV light.

Point 5: I do not understand this result, usually it is coconut agar is used

Response 5: I think PDA can also be reliably used.

Point 6: I have difficulties in accepting this result

Response 6: I think this results corresponds with what has been reported elsewhere. Aflatoxigenic isolates use their energies in Aflatoxin production and so do not grow as fast as non Aflatoxigenic isolates.

Point 7: I am not sure if this is relevant

Response 7: I think this is relevant because it will help determine which of the isolates grows faster and help address the point 6 raised above.

Point 8:  A different result that will be a novelty. The result is exactly what everyone is expecting

Response 8: Thanks for your comments. I agree that, that is everyone expectations. I also think it could also be possible we may not get what everyone is expecting. The HPLC helped to confirm our earlier findings about the toxigenicity of the isolates using cultural methods of detection.

Point 9:  It is already said according to manufacturers guidelines

Response 9: This has been corrected as suggested for all subsequent scenarios

Point 10:  Italicize scientific names

Response 10: This has has been done for all.

Point 11: No microscopic identification, I am not convinced you have identified correctly the isolates

Response 11: Microscopically Aspergillus species have same spore morphology. But A. flavus can readily be distinguished morphologically by the production of a bright yellow-green conidial color when cultured on PDA. This can be detected macroscopically.

Point 12:  Why acetic acid? According to the method that you mentioned, they used methonol and acetronitrile. Usually acetic acid is used in ochratoxin extraction.

Response 12: The modification of the extraction solution to acetonitrile: acetic acid (9:1) as it results in a better separation of the organic phase from the aqueous phase and sample after centrifugation.

Point 13:  No acetronitrile? With this mobile phase, I would be expecting longer run times and retention times.

Response 13: We observed that using a column of length 30cm resulted in a longer run time and peak tailing, especially for the Aflatoxin B1 peak

Reviewer 2 Report

The article "Growth and toxigenicity of A. flavus on resistant and susceptible peanut genotypes" presents the isolation and classification of toxigenic and non-toxigenic strains of Aspergillus flavus on peanuts and an in-vitro infectivity study on different seeds (resistant or non-resistant).

Although the work is complete and detailed, the search for non-toxigenic strains of mycotoxin-producing fungi is a research topic already being widely covered. Although it is valuable information for future trials, it lacks originality in its approach.

The results obtained open the possibility of using resistant varieties to control the incidence of A. flavus in peanuts, although further studies are needed to confirm these hypotheses.

It is recommended that the authors review the text in detail since significant grammatical errors and errors in the references to the figures have been detected.

Specific comments/remarks:

-Line 8: The first time A. flavus is named, the genus (Aspergillus) has to be indicated.

Line 19: Double space at the beginning of the sentence?

-Line 22: Non-aflatoxigenic.

-Line 43: Missing comma: Among the Aspergillus species, (...)

-Line 45: However, (...)

-Line 50: Remove comma

Line 72: Colony surface characteristics (...) are represented...

Line 76: after 72 h of incubation

Line 79: Indicate the figure in parentheses: (Figure 1)

Line 86, 92, 99, 102...: Figure references are poorly indicated, which makes it difficult to understand the work.

Table 1: The presentation of the Table should be improved.

-The figures presented each have a different format; it is recommended that the authors use a single model figure to facilitate understanding.

-Similarly, in the text, there is information related to statistical analysis that is not present in the figures. It is recommended that the authors add the appropriate information to the figures to confirm what is indicated in the text.

- The figure captions are not very explanatory. It is recommended that the authors incorporate sufficient information so that it is not necessary to refer to the text to understand the figures.

-The discussion is correct and adequately referenced. However, it would be interesting to include in this section studies that have had the same objective as the one proposed by the authors, to search for non-toxigenic strains of A. flavus or mycotoxin-producing fungi as a strategy to reduce the population of toxigenic fungi.

-The conclusions are adequate according to the objectives of the study.

-In Materials and methods, most of the titles (5.2.3, 5.2.5, 5.2.7...) A. flavus is not italicized.

- Line 300: double spacing

- Line 303, 311and 321: consistency in the way temperature units are represented (one space is missing).

- Line 337: A. flavus

- In the infection study, an aflatoxin-producing strain and a non-producing strain were used. However, it is not indicated whether it is a particular isolate that has been previously performed. Indicate in the text.

-HPLC quantification of aflatoxins, has the matrix effect been taken into account?

Author Response

-Line 8: The first time A. flavus is named, the genus (Aspergillus) has to be indicated.----

Response

This has been corrected.

Line 19: Double space at the beginning of the sentence? ---

Response

This has been resolved

-Line 22: Non-aflatoxigenic.-----

Response

This has been resolved

-Line 43: Missing comma: Among the Aspergillus species, (...)----

Response

This has been resolved

-Line 45: However, (...)------

Response

This has been resolved. But however has rather been changed to although.

Response

-Line 50: Remove comma-----

This has been corrected.

Line 72: Colony surface characteristics (...) are represented... ----

Response

Reviewer 1 thinks that figure was not needed and so it has been deleted

Line 76: after 72 h of incubation----

Response

This entire paragraph has now been deleted based on reviewer 1 comments.

Line 79: Indicate the figure in parentheses: (Figure 1) -----

Response

Same comment as above applies

Line 86, 92, 99, 102...: Figure references are poorly indicated, which makes it difficult to understand the work.

Response

This has been improved.

Table 1: The presentation of the Table should be improved.-----

Response

I am thinking the creation of pdf version caused some shifts in the original table. This has been corrected now.

-The figures presented each have a different format; it is recommended that the authors use a single model figure to facilitate understanding.-----

Response

This has been resolved now

-Similarly, in the text, there is information related to statistical analysis that is not present in the figures. It is recommended that the authors add the appropriate information to the figures to confirm what is indicated in the text. -------

Response

This has been rectified.

The figure captions are not very explanatory. It is recommended that the authors incorporate sufficient information so that it is not necessary to refer to the text to understand the figures.----

Response

This has been improved now.

-The discussion is correct and adequately referenced. However, it would be interesting to include in this section studies that have had the same objective as the one proposed by the authors, to search for non-toxigenic strains of A. flavus or mycotoxin-producing fungi as a strategy to reduce the population of toxigenic fungi. ----

Response

Some information have now been provided for this backed with references.

-The conclusions are adequate according to the objectives of the study.-----

Response

Thank you

-In Materials and methods, most of the titles (5.2.3, 5.2.5, 5.2.7...) A. flavus is not italicized.-----

Response

This has been resolved now

Line 300: double spacing ----

Response

This has been resolved

Line 303, 311 and 321: consistency in the way temperature units are represented (one space is missing)

Response

- Line 337: A. flavus

In the infection study, an aflatoxin-producing strain and a non-producing strain were used. However, it is not indicated whether it is a particular isolate that has been previously performed. Indicate in the text.---

Response

This has now been indicated at the materials and methods section

-HPLC quantification of aflatoxins, has the matrix effect been taken into account?

Response

---- Yes, we used matrix-based calibration for the study (This have been indicated it in the HPLC section)

Reviewer 3 Report

The present paper was poorly written. It has shortcomings in structure and lacks coherency. The result section especially was inadequately described. I suggest that the authors thoroughly work on the manuscript to correct the flaws. I observed deficiency in language making it difficult for the reader to understand.

Comments

According to the journal format, “Abstract: The abstract should be a total of about 200 words maximum”. The author failed to adhere to this format.

Use the correct citation style

Line 41 – 46, authors should write in a coherent manner. Restructure the sentences and avoid repetition

Line 47- 49, consequence in peanut? Correct sentences

Line 50 – 51. Repetition

Line 67 – 70, correct sentence

Line 72, what lines? Clarify

Line 86, 92 Error! Reference source not found? Correct this error although the manuscript

Line 126 – 129, correct sentence

Line 135 - 140, correct sentence

Line 137, specify the resistant genotype?

Line 138, What is the highest? State the value

Line 153-155, correct sentence

Line 156 – 159, difficult to understand

Line 160-162, put data in a table format

Line 163, Indicates…. Are you referring to the figure below? Clarify

Line 176, add reference

Line 195, 207, 210, …made by who? Check citation style and correct accordingly. Effect correction throughout the manuscript

Line 202, … media have different abilities in detecting aflatoxigenic and non-aflatoxigenic A. flavus isolates? Correct sentence

Line 251 – 254, correct sentence  

Line 297, what the peanuts infected before purchase? What is the source of the fungi.

Figure 2. on what medium are figures 2A and 2C? the figure description is not clear. Correct accordingly

Figure 4. What is A8, A2, NA11, NA12? Describe in the title of the figure for clarity

Figure 5. what is AUDPC. Define

Figure 6 and 7, specify the resistant genotype

Author Response

According to the journal format, “Abstract: The abstract should be a total of about 200 words maximum”. The author failed to adhere to this format.----- The abstract has now been reduced to less than 200 words as follows:

Aflatoxin contamination poses serious health concerns to consumers of peanut and peanut products. This study aimed at investigating the response of peanuts to Aspergillus flavus infection and aflatoxin accumulation. Isolates of A. flavus were characterized either as aflatoxigenic or non-aflatoxigenic using multiple cultural techniques. The selected isolates were used in an in-vitro seed colonisation (IVSC) experiment on two A. flavus resistant and susceptible peanut genotypes. Disease incidence, severity and aflatoxin accumulation were measured. Genotypes differed significantly (P < 0.001) in terms of the incidence and severity of aflatoxigenic and non-aflatoxigenic A. flavus infection with the non-aflatoxigenic isolate having significantly higher incidence and severity values. There was no accumulation of aflatoxins in peanut genotypes inoculated with non-aflatoxigenic isolate indicating its potential as a biocontrol agent. Inoculations with the afla-toxigenic isolate resulted in the accumulation of aflatoxin B1 and G1 in all the peanut genotypes. Aflatoxin B2 was not detected in ICGV-03401(resistant genotype) while it was present and higher in Manipinta (susceptible genotype) than L027B (resistant genotype). ICGV-03401 can resist fungal infection and aflatoxin accumulation than L027B and Manipinta. Non-aflatoxigenic isolate detected in this study could further be investigated as a biocontrol agent.

Use the correct citation style

Line 41 – 46, authors should write in a coherent manner. Restructure the sentences and avoid repetition—This has been corrected as follows.

Peanut production is faced with challenges such as inadequate inputs, unreliable rains, the use of unimproved varieties, disease and pest infestation [9]. In addition to the above, peanut is reported to be prone to Aspergillus species infection [10–12] which leads to the decline in its quality and quantity [13,14]. Among the Aspergillus species, A. flavus and A. parasiticus are the major contributors to aflatoxin production at the pre and postharvest stages although, A. flavus is reported to be the most prevalent in Ghana [14, 15]. Aflatoxin contamination is a food safety concern due to its detrimental health impact and its resulting economic loss to peanut growers and dealers [16]. Aflatoxin contamination can also result in the reduction in the quality of grain and oilseed and depletion in their nutritional value [11].

Line 47- 49, consequence in peanut? Correct sentences--- This has been corrected

Line 50 – 51. Repetition---- This has been corrected as follows

Research discoveries have established that A. flavus is the most dominant and the most frequently isolated species from peanut seeds, products and farmers’ fields leading to aflatoxin contaminations [13, 17–20]

Line 67 – 70, correct sentence---This has been corrected as follows

This study seeks to culturally characterize A. flavus isolates based on their aflatoxigenicity and distinguish resistance mechanisms acting on resistant and susceptible peanut genotypes following inoculations with aflatoxigenic and non-aflatoxigenic A. flavus in-vitro.

Line 72, what lines? Clarify--- This has been corrected as follows;

Measurements of fungal incidence, severity and aflatoxin accumulations in the A. flavus resistant peanut genotypes (ICGV03401 and L027B) and susceptible peanut genotype (Manipinta ) will help to distinguish the different resistance mechanisms acting in them.

Line 86, 92 Error! Reference source not found? Correct this error although the manuscript---Sorry this has been corrected

Line 126 – 129, correct sentence--- This has been corrected.

Also, the area under the disease progress curve (AUDPC) values of aflatoxigenic A. flavus isolates (A2 and A8) were significantly lower than AUDPC values for non-aflatoxigenic A. flavus isolates (NA11 and NA12) during the 8 days incubation period (Figure 4).

Line 135 - 140, correct sentence---- This has been corrected.

There was highly significant (P < 0.001) variation in incidence of the three peanut genotypes (ICGV03401, L027B and Manipinta) inoculated with aflatoxigenic and non-aflatoxigenic A. flavus. The resistant genotypes (ICGV03401 and L027B) had incidence values less than 40% for aflatoxigenic and non-aflatoxigenic A. flavus whereas the susceptible genotype (Manipinta) had the highest (78%-80%). However, the percentage incidence of non-aflatoxigenic A. flavus on all the genotypes was higher than the percentage incidence of aflatoxigenic A. flavus (Figure 5). 

Line 137, specify the resistant genotype? ---- This has been corrected

Line 138, What is the highest? State the value This has been corrected …. A. flavus whereas susceptible genotype (Manipinta) had the highest (53-63%).

Line 153-155, correct sentence----- This has been corrected as follows  --Extraction and estimation of aflatoxin accumulation in aflatoxigenic and non-aflatoxigenic isolates in peanut was quantified using HPLC (Figure 7) with aflatoxin standards (Figure 8).

Line 156 – 159, difficult to understand----- This has been corrected

Line 160-162, put data in a table format--- This has been resolved

Line 163, Indicates…. Are you referring to the figure below? Clarify—This has been resolved

Line 176, add reference

Line 195, 207, 210, …made by who? Check citation style and correct accordingly. Effect correction throughout the manuscript--- This has been corrected

Line 202, … media have different abilities in detecting aflatoxigenic and non-aflatoxigenic A. flavus isolates? Correct sentence---- This has been corrected

Line 251 – 254, correct sentence  --- This has been corrected

Line 297, what the peanuts infected before purchase? What is the source of the fungi--- This has been corrected

Figure 2. on what medium are figures 2A and 2C? the figure description is not clear. Correct accordingly---This has been addressed.

Figure 4. What is A8, A2, NA11, NA12? Describe in the title of the figure for clarity—This has been addressed.

Figure 5. what is AUDPC. Define A---- Area under the disease progress curve—Has been resolved now.

Figure 6 and 7, specify the resistant genotype--- This has been resolved

Reviewer 4 Report

The manuscript describes the the response of peanuts to A. flavus infection and aflatoxin accumulation exploiting the in-vitro seed colonisation.

In my opinion the manuscript would be of interest for the Toxins readership and needs just some minor revisions before publication.

Below some suggestions to further improve the manuscript.

Introduction.

Page 1, line 36. I am not sure if it is necessary to specify “especially women”.

Results.

Page 3, line 86. Please check “Error! Reference source not found..”. Please revise it in the whole manuscript.

Page 3, lines 88,89 “A represents A. flavus blue fluorescent colour and B did not fluoresce on PDA.” Please revise the sentence.

Page 4, lines 97,98. Please revise the percentages.

Page 8, line 163. Please revise “absents”.

Materials and methods.

Page 13 line 337. Please specify y and t.

Page 14 line 387. Please convert rpm in g.

Author Response

Introduction.

Page 1, line 36. I am not sure if it is necessary to specify “especially women”. ----- Especially women has been removed

Results.

Page 3, line 86. Please check “Error! Reference source not found..”. Please revise it in the whole manuscript.-------------- This has been corrected

Page 3, lines 88,89 “A represents A. flavus blue fluorescent colour and B did not fluoresce on PDA.” Please revise the sentence.---- Thanks very much. We have corrected this.

Page 4, lines 97,98. Please revise the percentages. ----Please, I did not get this part very well. But I believe the current revision may have improved this.

Page 8, line 163. Please revise “absents”. -----This has been revised.

Materials and methods.

Page 13 line 337. Please specify y and t. ---- This has now been indicated.

Page 14 line 387. Please convert rpm in g. ----- This is now 3300 x g

Round 2

Reviewer 1 Report

Dear Authors,

Thank you for you effort in answering my questions, however there are still some issues that need to be addressed. I am attaching the manuscript along with you answer where you can find more comments. I hope to see you work improved.

Regards

Reviewer 2 Report

Thanks for doing the corrections according to the reviewers suggestions. The quality of the manuscript has increased.

Author Response

Thanks very much for your time in reviewing this manuscript. It has really helped to improve the quality.